# Movement Behaviors and Perceived Loneliness and Sadness within Alaskan Adolescents

**DOI:** 10.3390/ijerph17186866

**Published:** 2020-09-20

**Authors:** Ryan D. Burns, Yang Bai, Christopher D. Pfledderer, Timothy A. Brusseau, Wonwoo Byun

**Affiliations:** Department of Health & Kinesiology, College of Health, University of Utah, Salt Lake City, UT 84112, USA; Yang.Bai@utah.edu (Y.B.); chris.pfledderer@utah.edu (C.D.P.); tim.brusseau@utah.edu (T.A.B.); won.byun@utah.edu (W.B.)

**Keywords:** physical activity, sadness, sedentary behavior, sleep, survey

## Abstract

Physical activity, screen use, and sleep are behaviors that integrate across the whole day. However, the accumulative influence of meeting recommendations for these 24-h movement behaviors on the mental health of Alaskan adolescents has not been examined. The purpose of this study was to examine the associations between movement behaviors, loneliness, and sadness within Alaskan adolescents. Data were obtained from the 2019 Alaska Youth Risk Behavior Survey (YRBS). The number of adolescents participating in the 2019 Alaska YRBS was 1897. Associations between meeting recommendations for movement behaviors with loneliness and sadness were examined using weighted logistic regression models, adjusted for age, sex, race/ethnicity, and body mass index (BMI). Approximately 5.0% of the sample met recommendations for all three movement behaviors. Meeting 2 or 3 movement behavior recommendations was associated with lower odds of loneliness (odds ratio (OR) range = 0.23 to 0.44, *p* < 0.01). Additionally, meeting 1 to 3 movement behavior recommendations was associated with lower odds of sadness (OR range = 0.29 to 0.52, *p* < 0.05). Joint association analyses determined that these relationships were primarily driven by meeting the sleep recommendation for loneliness and meeting the screen use recommendation for sadness. The results support use of multiple movement-based behavior programming to attenuate feelings of loneliness and sadness within Alaskan adolescents.

## 1. Introduction

The analysis of 24-hour movement behaviors has recently shown that physical activity, sedentary behaviors, and sleep are associated with a number of health outcomes within the pediatric population [1,2,3,4]. The 24-hour movement behavior recommendations are relevant to the pediatric population regardless of sex, race/ethnicity, or socioeconomic status [1,2]. An optimal composition of these movement behaviors includes achieving at least 60 min of physical activity per day including at least 3 days per week of muscle strengthening activities, limiting screen use to 2 h or less per day, and receiving an adequate amount of sleep (e.g., 8–10 h per night for adolescents aged 14 years old or older) [1]. Because many populations of youth do not meet recommendations for these movement behaviors [1,2], the examination of these behaviors concurrently using various analytic approaches is of importance to yield new information that can be applied within school- and community-based settings. Indeed, novel analytic approaches such as compositional data analysis have shown that the 24-hour composition of movement behaviors correlate with cardiometabolic risk, health-related fitness, weight status, and academic performance outcomes [5,6,7,8]. However, the relationships between the additive influences of movement behaviors on mental health outcomes is not as well researched [2]. 

Given the current global COVID-19 pandemic and the associated social/physical distancing recommendations, mental health is a significant public health concern now more than ever within the pediatric population. Elevated levels of loneliness and sadness can have detrimental effects on the health and well-being of youth [9,10]. Adolescents with higher levels of anxiety and depression resulting from loneliness and sadness are more likely to have these mental health conditions in adulthood, where it can negatively affect work production and overall health and well-being [11,12]. Adolescents with poor mental health are also more likely to have poorer academic performance [13] and increased risk of suicide [14,15]. Pfledderer et al. [16] found that meeting physical activity guidelines and hours of sleep per night significantly predicted suicidal ideation in a national sample of U.S. adolescents. Using data from the Ontario Child Health Study, Kim et al. [17] found that adolescents reporting 4 or more hours of passive screen use time per day, compared to those reporting less than 2 h per day, were three times more likely to meet criteria for a major depressive episode, social phobia, and generalized anxiety disorder. However, active screen time did not significantly associate with mental health disorders. In a large longitudinal study, Boers et al. [18] found that for every 1 hour spent on social media and general computer use, adolescents showed a significant increase in depressive symptoms. The authors concluded that these time-varying associations were partially explained by upward social comparisons. These studies suggest that individually, higher levels of physical activity, low levels of sedentary behaviors such as television watching and non-academic computer use, and adequate levels of school-night sleep have been shown to associate with lower suicidal ideation [16], anxiety [17], and depressive symptoms in adolescents [18]. However, only a few studies have thoroughly examined the accumulative influences of these movement behaviors within specific populations of youth [19,20]. Meeting recommendations for all 24-hour movement behaviors may be associated with a significantly lower probability of poor mental health compared to that when just meeting one movement recommendation. 

Within the U.S., much research has focused on health behaviors and health outcomes in youth residing within the lower 48 states. Adolescents within the U.S. state of Alaska have not been as well researched, especially on the links between movement behaviors and mental health outcomes [21]. Alaskan adolescents reside in unique climates that may not be conducive to achieving recommended levels of physical activity, screen time, and school-night sleep. Temperatures in Alaska tend to be colder than temperatures observed within the lower 48 states, especially within the sub-artic and arctic climate regions, and Alaskan daylight hours during the winter months are fewer due to its geographical location within the northern hemisphere [22]. Therefore, patterns of movement-based behaviors of Alaskan youth may be different compared to those of youth residing within the lower 48 states. The Alaskan demographic is also characterized by having a higher proportion of youths who are of American Indian and Alaskan Native race/ethnicity [23]. These youths have been found to have a higher prevalence of mental health issues compared to youth within the general U.S. population [21]. The examination of movement behaviors within this specific pediatric population is lacking. Furthermore, feelings such as loneliness and sadness can contribute to mental health issues such as depression, personality disorders, and Alzheimer’s disease [24,25]. Sadness tends to share the same clinical correlates as major depressive disorder [25]. Examining the association between movement-based behaviors and these mental states can provide important evidence for potential protective effects within a unique sample of Alaskan youth that can be used in future school- and community-based intervention efforts. Therefore, the purpose of this study was to examine the associations of movement-based behaviors with loneliness and sadness within a sample of Alaskan adolescents.

## 2. Materials and Methods

### 2.1. Participants

The sample’s demographic characteristics are presented in Table 1. The average age of the sample was 15.8 ± 1.3 years old. The sex distribution was approximately homogenous with the majority of the sample being non-Hispanic Whites. American Indians and Alaskan Natives comprised of nearly one-quarter of the sample, with a weighted prevalence of nearly one-third of the sample. Based on self-reported height and weight, the average body mass index (BMI) was 23.6 ± 5.2 kg/m^2^. Using the 2000 Centers for Disease Control and Prevention (CDC) BMI-for-age growth charts, the average BMI z-score was 0.61 ± 0.96. 

### 2.2. Instrumentation

The Alaska Youth Risk Behavior Survey (YRBS) is part of a data collection system established by the Centers for Disease Control and Prevention (CDC) in 1990, which was first implemented in Alaska in 1995. The YRBS includes questions on current health and risk behaviors such as physical activity; nutrition; tobacco, alcohol, and drug use; safety, violence, and bullying; suicide; sexual behaviors; and connections with peers, adults, and the community. The Alaska YRBS is an anonymous and voluntary survey of students in grades 9–12 in public schools and is administered throughout the state of Alaska in odd-numbered years. The 2019 Alaska YRBS response rates yielded 39 out of 43 (91%) sampled traditional high schools and 1897 of 2824 (67%) sampled traditional high school students. The overall response rate was 61% (91% × 67%). The procedures for collecting YRBS data on human subjects were approved by CDC’s Institutional Review Board (#1969.0). Parental written consent was obtained prior to data collection on a respective Alaskan adolescent [23].

### 2.3. Procedure

The Alaska statewide traditional high school sample includes students in traditional public high schools with an enrollment of at least 10 students. The sample excluded boarding, correspondence, home study, alternative, and correctional schools. Adolescents were selected using a two-stage cluster sample design. The first stage consisted of selecting schools with a probability proportional to the school enrollment size. The second stage consisted of randomly selecting classes within each school. The Alaska YRBS statewide data are weighted by sex within race/ethnicity. Alaskan YRBS data are representative of all Alaska traditional high school students [23]. 

There were two separate outcome variables within the current study consisting of perceived loneliness and prolonged sadness. The perceived loneliness item stated, “I feel alone in my life” with 5-point Likert-type responses ranging from “strongly agree” to “strongly disagree”. The perceived loneliness item was recoded as 0 = not sure, disagree, and strongly disagree and 1 = agree and strongly agree for analysis. The prolonged sadness item asked, “During the past 12 months, ever feel so sad for two weeks or more and stop usual activities?” with a binary yes/no response. The prolonged sadness item was recoded as 0 = no and 1 = yes for analysis. Four binary coded movement-based behaviors were the predictor variables, specifically derived from items asking about the weekly frequency of 60 min of physical activity per day, weekly frequency of muscle strengthening activity, hours per day of television watching and non-academic computer screen use, and hours of school-night sleep. The physical activity binary predictor was derived by combining the 60 min of physical activity per day item and the muscle strengthening item (0 = did not meet both recommendations, 1 = met both recommendations, i.e., 60 min per day of physical activity and 3 days per week or more of muscle strengthening). The screen use binary predictor variable was derived by combining the television and computer use predictors so that 0 = more than 2 h of combined screen use per day and 1 = 2 h or less per day. The sleep duration binary predictor was coded as 0 = less than 8 h per school night and 1 = 8 or more hours per school night. These cut-points were based off of current 24-hour movement-based recommendations for physical activity, screen use, and sleep [1,2]. 

### 2.4. Statistical Analysis

The complex YRBS survey design, including assigned stratum and primary sampling unit, was accounted for using Stata’s “svyset” prefix command. Missing data were not imputed. Weighted analyses used the Taylor Series Linearization variance estimation. For all categorical variables, unweighted and weighted prevalence statistics were reported. To examine the associations between each movement-based behavior and perceived loneliness and prolonged sadness, weighted logistic regression models were employed. Separate models were employed for each outcome variable. Crude and adjusted parameter estimates (odds ratios) with 95% confidence intervals (CIs) were calculated and reported. Within the adjusted models, age, sex, race/ethnicity, and BMI z-scores were included as covariates to adjust for potential confounding influences. 

Additional analyses consisted of examining the relationship between meeting accumulative (0 to 3) healthy movement-based behavior recommendations and the odds of perceived loneliness and prolonged sadness. A healthy movement-based behavior composite variable was derived by adding intraindividual counts of meeting recommendations of movement-based behaviors (i.e., physical activity, screen use, sleep). A weighted logistic regression model was then run using the derived composite variable as the primary predictor, adjusting for age, sex, race/ethnicity, and BMI z-scores. The reference level for comparison was meeting 0 movement-based recommendations. To determine specific joint associations, a final analysis was employed testing possible two-way and three-way interactions on both perceived loneliness and prolonged sadness. Meeting no recommendations was again used as the reference level. All analyses had an alpha level of *p* < 0.05 and were carried out using Stata v15.0 statistical software package (Statacorp, College Station, TX, USA). 

## 3. Results

Table 2 presents the descriptive statistics for perceived loneliness, prolonged sadness, and movement-based behaviors. Approximately 20.9% of the sample did not meet any movement recommendations, 49.7% met 1 recommendation, 24.4% met 2 recommendations, and 5.0% of the sample met all 3 movement recommendations. The relationships between movement-based behaviors, perceived loneliness, and prolonged sadness are presented in Table 3 and Table 4, respectively. Within the adjusted models, sleeping for at least 8 h per school night was significantly associated with lower odds of perceived loneliness (OR = 0.43, 95% CI: 0.24–0.75, *p* = 0.004) and lower odds of prolonged sadness (OR = 0.48, 95% CI: 0.33–0.70, *p* < 0.001). Physical activity and screen use were not significantly associated with either outcome within the adjusted models. 

Figure 1 presents the association between the movement behavior aggregate variable (high physical activity and muscle strengthening, low screen use, adequate school-night sleep duration) with perceived loneliness and prolonged sadness. Meeting recommendations for 2 (OR = 0.44, 95% CI: 0.24–0.81, *p* = 0.008) or 3 (OR = 0.23, 95% CI: 0.08–0.64, *p* = 0.005) movement-based behaviors was significantly associated with lower odds of perceived loneliness compared to adolescents who did not meet any recommendations. Meeting recommendations for 1 (OR = 0.52, 95% CI: 0.35–0.77, *p* = 0.002), 2 (OR = 0.37, 95% CI: 0.26–0.75, *p* < 0.001), or 3 (OR = 0.29, 95% CI: 0.11–0.77, *p* = 0.014) movement-based behaviors was significantly associated with lower odds of prolonged sadness compared to adolescents who did not meet any recommendations, with point estimates displaying an inverse dose–response association. 

Table 5 presents the joint associations for predicting perceived loneliness and prolonged sadness. All joint effects including meeting sleep duration recommendations significantly predicted perceived loneliness (*p* < 0.05). All joint effects including meeting screen use recommendations significantly predicted prolonged sadness (*p* < 0.05). 

## 4. Discussion

The purpose of this study was to examine the associations of movement-based behaviors with perceived loneliness and prolonged sadness within a sample of Alaskan adolescents. The results indicated that school-night sleep duration was the only single movement behavior that was significantly associated with perceived loneliness and prolonged sadness. However, the derived composite movement behavior predictor variable was significantly associated with both perceived loneliness and prolonged sadness in a dose-dependent manner. Joint association analyses determined that the accumulative relationships were primarily driven by meeting the sleep recommendation for loneliness and meeting the screen use and sleep recommendations for sadness. The results indicate that there may be an additive effect of these movement behaviors that when considered together can serve as protection against perceived loneliness and prolonged sadness in Alaskan adolescents. Interpretations of these findings and implications for Alaskan school and community health programming are provided further. 

A primary finding from the current study was that sleep was the only single movement-based behavior that was significantly associated with both perceived loneliness and prolonged sadness in Alaskan adolescents. Studies have shown that both quantity and quality of sleep correlate with mental health outcomes in adolescents [26,27,28]. Interestingly, subjective sleep markers have been found to be more consistent and common across pediatric mental health disorders than objective sleep measures [29]. Proposed mechanisms as to why poor sleep is associated with mental health disturbances include reduced latency to Rapid Eye Movement (REM)sleep, high sleep onset latency, and a high number of sleep arousals [29,30,31]. Adolescents with depression have showed higher peri-sleep onset of the stress hormone cortisol [32]. Children and adolescents with anxiety show higher bedtime fears and greater wake after sleep onset durations [30,31,32]. Although the items used in the current study did not specifically examine anxiety and depression, loneliness and sadness may be associated with these mental health disorders [33,34], making it plausible that similar physiological and behavioral mechanisms are involved in the association between sleep, loneliness, and sadness. 

Even though school-night sleep duration significantly correlated with loneliness and sadness, the movement behaviors of physical activity and screen use did not correlate within any of the adjusted models. In a longitudinal study using a cohort of 928, 12- to 13-year-old youths from England, no significant association was found between objectively measured physical activity at baseline and mental well-being assessed 3 years later [35]. Additionally, a recent review has indicated that physical activity may have significant but small associations with anxiety in adolescents [36]. A review of longitudinal studies examining the association between physical activity and loneliness yielded mixed findings, with only a few studies showing a direct inverse relationship between the two constructs and other studies showing indirect effects with mediated mechanisms [37]. Within the current study, physical activity was associated with loneliness and sadness within the crude models, but after covariate adjustment, these significant associations were not present. At least within the general Alaskan adolescent population, self-reported physical activity does not seem to be a significant correlate of loneliness and sadness. 

The prevalence of meeting screen use recommendations in the current sample of Alaskan adolescents is much higher (>50%) compared to that among other population-based studies. For example, using the Canadian COMPASS survey, only 6.4% of students within grade 9 met screen use recommendations [38]. It is unclear why the prevalence of meeting screen use recommendations in Alaskan adolescents is so much higher than that among other populations. This may be due to a relative lack of access to television and/or computer screens/devices within the home environment, possibly related to socioeconomic status, although this information is not collected on the YRBS. It may also be associated with cultural differences, especially within the American Indian and Alaskan Native population. Nevertheless, another recent Alaskan descriptive survey indicated that adolescent screen use during COVID-19 is drastically higher than that before COVID-19 [39]. Even though the prevalence of meeting screen use recommendations from the 2019 Alaskan YRBS is relatively high, excessive screen use in this population is still a major concern, especially within the current COVID-19 climate.

Again, concerning screen use, the null findings relating screen use with mental health outcomes from the current study do conflict with past survey-based research. Using the parent-reported National Survey of Children’s Health, it was found that more than 1 hour of screen use per day was associated with lower psychological well-being, including less curiosity, lower self-control, more distractibility, more difficulty making friends, less emotional stability, being more difficult to care for, and inability to finish tasks among a large representative sample of children and adolescents [40]. Additionally, other authors have observed time-varying associations between social media use, television watching, and depression in adolescents [18]. Data from the Global School-Based Student Health Survey indicates that being sedentary for more than 8 h per day was associated with a twice-as-high risk for feeling lonely among a large sample of adolescents [41]. Within the current study, even though television watching, and computer use did not significantly associate with either loneliness or sadness within the adjusted models, point estimates do suggest perhaps some protective effect. The lack of statistical significance may be due to large variability within the Alaskan sample. This variability may be due to distinct cultural differences among the racial/ethnic groups within the Alaskan population, large variation in socioeconomic status within the Alaskan population, and/or possibly, self-report measurement issues. Because the YRBS is self-report based and social desirability bias is possible, a larger sample size may yield statistical associations between screen use variables, loneliness, and sadness. An important secondary finding within the adjusted model was that American Indian and Alaska Native adolescents had higher odds of reporting prolonged sadness compared to non-Hispanic White adolescents. This specific finding is in accordance with information presented in other work and stresses the importance of deriving potential future multi-behavioral health programming within this pediatric population [21]. 

The most salient finding from the current study was that a movement-based behavior composite variable correlated with both perceived loneliness and prolonged sadness in a dose-dependent manner. Joint association analyses determined that these relationships were primarily driven by meeting sleep recommendations for loneliness and meeting screen use recommendations for sadness. Other studies have shown that movement-based behaviors may have additive effects on mental health outcomes within the pediatric population. Zhu et al. [19] found that meeting recommendations for physical activity, screen use, and sleep yielded lower odds of anxiety in children and lower odds of depression in adolescents using the parent-reported Nation Survey of Children’s Health. Sadness and depression are closely linked; therefore, the current study’s findings support this previous work. Ogawa et al. [20] found significant joint effects between physical activity and sleep on anxiety and depression using an adolescent sample of Japanese adolescents. Using data from the U.K.-Millennium Cohort Study, Pearson et al. [42] found that only 9.7% of the sample met concurrent recommendations for physical activity, screen use, and sleep. Pearson et al. [42] also found that adolescent boys who were obese and those with depressive symptoms were significantly less likely to meet all three movement recommendations. Within the current study, the aggregate movement behavior variable correlated more strongly with prolonged sadness than with perceived loneliness. This finding is in accordance with the aforementioned previous work, assuming self-reported prolonged sadness and depression are related. The current study also provides some evidence of a dose–response mechanism, although only a small proportion of the Alaskan adolescent sample met all three movement recommendations (approximately 5%), making it difficult to draw definitive conclusions regarding the form of the relationship. 

It is concerning that over a quarter of adolescents who took part in this study reported feelings of perceived loneliness and even more reported feelings of prolonged sadness. Both loneliness and prolonged sadness have been linked to more serious issues in adolescents, including anxiety, depression, and suicidal ideation. In a recent meta-analysis, Maes et al. [43] found a strong, positive cross-sectional association between loneliness and social anxiety and also found that this relationship held over time when considering the associations longitudinally. Another recent meta-analysis found that loneliness was a significant predictor of suicidal ideation/behavior and that being aged 16–20 years increased the likelihood that the relationship between loneliness and suicidal ideation would be significant [44]. Having feelings of sadness has also been linked to myriad risk behaviors among adolescents including binge drinking, gang membership, suicidal ideation, and suicide attempt [45]. While our study did not investigate these other risk behaviors, it is important to understand the wide range of implications that perceived loneliness and/or prolonged sadness might have on Alaskan youth. Additionally, as more schools transition from face-to-face instruction to online formats during the current COVID-19 pandemic, it is imperative that youth who already experience feelings of loneliness and/or prolonged sadness receive the support they need.

The findings also have important implications for schools. During adolescence, circadian rhythms change, which leads to later sleep and wake times [46]. As a result, the American Academy of Sleep Medicine recommends school starts times of 8:30 a.m. or later [47], which is supported by a growing body of literature highlighting how later school start times are correlated to increases in sleep times [48]. Beets and colleagues [49] suggest that most physical activity interventions that are successful fall into one (or more) of three categories: providing new opportunities for youth to be active (expansion), increasing the time available for youth to be active (extension), or improving the opportunity beyond what is typically offered (enhancement). More specifically, schools and community programs should focus on providing additional or expanded physical activity opportunities for youth that are innovative, well equipped, and of interest to the participants are more likely to change/increase physical activity behaviors. Increasingly, screen time is an important component of daily life for many adolescents. Parents and youth must work together to identify rules, limits, and a monitoring system to change screen behaviors [50] to possibly mitigate detrimental effects on mental health.

Strengths of this study include use of a representative of Alaskan traditional high school students, the examination of accumulated movement behaviors, and examining two mental health states, loneliness and sadness, that are pertinent within the current COVID-19 social/physical distancing climate. Limitations to this study includes the use of a cross-sectional research design that precludes cause-and-effect inferences and manifests concerns regarding the true directionality of the observed associations. Additionally, this study used adolescent self-report for all variables that were collected; therefore, social desirability bias may have mitigated the validity of the responses. The use of crude assessment tools may also attenuate the validity of the findings. Third, all of the movement predictor variables were dichotomized for analysis, which may lead to loss of information from the original measurement scale. Fourth, the 24-hour movement guideline for sleep has an upper limit of 10 h per night for adolescents 14 years old and older. The 2019 YRBS does not assess specific hours of sleep over 10 h per night, precluding analysis of the influence of engaging in excessive sleep durations on loneliness and sadness. Fifth, it is likely that other variables, such as socioeconomic status, which was not collected on the YRBS, may be an important effect modifier. Finally, the 2019 Alaskan YRBS only asked respondents about physical activity frequency, not intensity, and only asked about sleep quantity, not quality, which may play a very important role in the association with adolescent loneliness and sadness. 

## 5. Conclusions

In conclusion, school-night sleep duration was significantly associated with perceived loneliness and prolonged sadness. Additionally, a movement behavior composite variable was significantly associated with perceived loneliness and prolonged sadness in a dose-dependent manner. These joint relationships were primarily driven by meeting sleep recommendations for loneliness and meeting screen use and sleep recommendations for sadness. These results communicate the importance of considering multiple movement behaviors when examining determinants of adolescent mental health. The findings also highlight the need to infuse multi-behavioral health programming for Alaskan adolescents, with particular cultural considerations for American Indian and Alaskan Native students. Meeting multiple movement behavior recommendations show promise for facilitating good mental health in Alaskan adolescents. 

## Figures and Tables

**Figure 1 ijerph-17-06866-f001:**
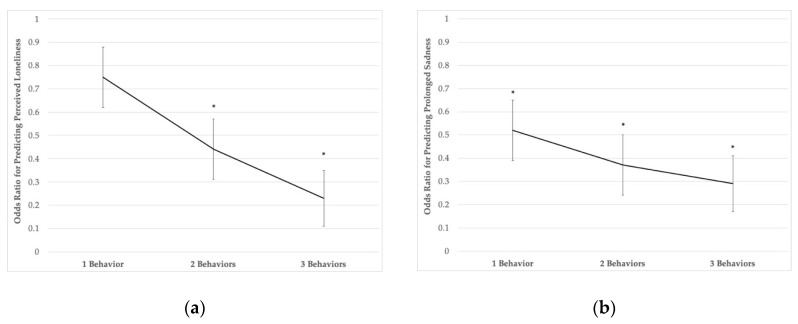
Odds ratios for predicting perceived loneliness (**a**) and prolonged sadness (**b**) across the number of met movement behavior recommendations. Referent level is meeting 0 movement recommendations; models adjusted for age, sex, race/ethnicity, and BMI z-score; * denotes statistical significance, *p* < 0.05.

**Table 1 ijerph-17-06866-t001:** Sample demographic characteristics.

Variable	Level	*n*	Unweighted Prevalence (%)	Weighted Prevalence (%)
Age	15 years old or younger	847	45.4%	38.8%
	16 or 17 years old	800	42.9%	45.2%
	18 years old or older	220	11.8%	16.0%
Sex	Male	919	49.7%	51.6%
	Female	930	50.3%	48.4%
Race/Ethnicity	White	921	50.6%	45.5%
	American Indian or Alaska Native	434	23.9%	31.4%
	Black or African American	49	2.7%	2.5%
	Hispanic or Latino	157	8.6%	5.4%
	Other Races	185	10.2%	11.4%
	Multiple Races	73	4.0%	3.9%
Weight Status	Healthy Weight	1164	67.0%	66.9%
	Overweight/Obese	572	33.0%	33.1%

**Table 2 ijerph-17-06866-t002:** Descriptive statistics for perceived loneliness, prolonged sadness, and movement behaviors.

Variable	Level	*n*	Unweighted Prevalence (%)	Weighted Prevalence (%)
Perceived Loneliness	No Perceived loneliness	1262	70.9%	72.7%
	Perceived loneliness	518	29.1%	27.3%
Prolonged Sadness	No Prolonged Sadness	1107	59.6%	61.9%
	Prolonged Sadness	749	40.4%	38.1%
Physical Activity Only	Recommendations Not Met	1474	81.2%	82.1%
	Recommendations Met	341	18.8%	17.9%
Physical Activity and Muscle Strengthening	Recommendations Not Met	1521	84.4%	85.5%
	Recommendations Met	281	15.6%	14.5%
Television and Computer Screen Use	2 h or more/Day	766	42.8%	39.8%
	Less than 2 h/Day	1024	57.2%	60.3%
School-Night Sleep Duration	Less than 8 h/Night	1352	75.4%	74.4%
	8 h/Night or More	441	24.6%	25.6%

**Table 3 ijerph-17-06866-t003:** Parameter estimates for predicting perceived loneliness.

Variable	Level	Crude OR (95% CI)	Adjusted OR ^1^ (95% CI)
Physical Activity and Muscle Strengthening	Recommendations Not Met	referent	referent
	Recommendations Met	0.66 * (0.45–0.98)	0.77 (0.52–1.17)
Television and Computer Screen Use	2 h or more/Day	referent	referent
	Less than 2 h/Day	0.74 * (0.56–0.98)	0.73 (0.52–1.02)
School-Night Sleep Duration	Less than 8 h/Night	referent	referent
	8 h/Night or more	0.47 * (0.28–0.78)	0.43 * (0.24–0.75)
Age (years)		0.99 (0.89–1.11)	1.02 (0.91–1.16)
BMI z-score		1.16 (0.99–1.37)	1.20 * (1.00–1.43)
Sex	Male	referent	referent
	Female	1.43 * (1.12–1.82)	1.44 * (1.11–1.85)
Race/Ethnicity	White	referent	referent
	American Indian or Alaska Native	1.04 (0.76–1.44)	1.05 (0.76–1.43)
	Black or African American	1.19 (0.53–2.64)	0.83 (0.34–1.98)
	Hispanic or Latino	1.34 (0.79–2.27)	1.21 (0.75–1.96)
	Other Races	1.56 * (1.04–2.34)	1.36 (0.88–2.11)
	Multiple Races	1.21 (0.68–2.15)	1.11 (0.61–2.03)

OR stands for odds ratio; 95% CI stands for 95% confidence interval; ^1^ OR adjusted for age, sex, race/ethnicity, and body mass index (BMI) z-score; * denotes statistical significance, *p* < 0.05.

**Table 4 ijerph-17-06866-t004:** Parameter estimates for predicting prolonged sadness.

Variable	Level	Crude OR (95% CI)	Adjusted OR ^1^ (95% CI)
Physical Activity and Muscle Strengthening	Recommendations Not Met	referent	referent
	Recommendations Met	0.65 * (0.45–0.95)	0.72 (0.48–1.08)
Television and Computer Screen Use	2 h or more/Day	referent	referent
	Less than 2 h/Day	0.83 (0.63–1.10)	0.74 (0.54–1.02)
School-Night Sleep Duration	Less than 8 h/Night	referent	referent
	8 h/Night or more	0.54 * (0.36–0.82)	0.48 * (0.33–0.70)
Age (years)		0.99 (0.88–1.12)	1.02 (0.91–1.16)
BMI z-score		0.92 (0.83–1.03)	0.91 (0.81–1.03)
Sex	Male	referent	referent
	Female	2.36 * (1.75–3.18)	2.24 * (1.61–3.13)
Race/Ethnicity	White	referent	referent
	American Indian or Alaska Native	1.47 * (1.02–2.07)	1.72 * (1.24–2.49)
	Black or African American	1.02 (0.45–2.33)	0.96 (0.44–2.10)
	Hispanic or Latino	1.57 * (1.06–2.33)	1.51 * (1.07–2.14)
	Other Races	1.10 (0.73–1.66)	1.08 (0.67–1.72)
	Multiple Races	1.50 (0.88–2.56)	1.56 (0.83–2.94)

OR stands for odds ratio; 95% CI stands for 95% confidence interval; **^1^** OR adjusted for age, sex, race/ethnicity, and BMI z-score; * denotes statistical significance, *p* < 0.05.

**Table 5 ijerph-17-06866-t005:** Joint associations for predicting perceived loneliness and prolonged sadness.

Outcome Variable	Recommendations Met	Adjusted OR ^1^ (95% CI)
Perceived Loneliness	All Behaviors Not Met	referent
	Screen Use and Sleep	0.32 * (0.13–0.77)
	Physical Activity and Sleep	0.17 * (0.04–0.71)
	Physical Activity and Screen Use	0.84 (0.42–1.66)
	Physical Activity, Screen Use, and Sleep	0.23 * (0.08–0.64)
Prolonged Sadness	All Behaviors Not Met	referent
	Screen Use and Sleep	0.37 * (0.24–0.57)
	Physical Activity and Sleep	0.47 (0.21–1.06)
	Physical Activity and Screen Use	0.56 * (0.33–0.91)
	Physical Activity, Screen Use, and Sleep	0.29 * (0.11–0.77)

Meeting physical activity recommendations includes muscle strengthening; OR stands for odds ratio; 95% CI stands for 95% confidence interval; **^1^** OR adjusted for age, sex, race/ethnicity, and BMI z-score; * denotes statistical significance, *p* < 0.05.

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
