# Peer review of "Movement Behaviors and Perceived Loneliness and Sadness within Alaskan Adolescents"

_ijerph, 2020, doi:10.3390/ijerph17186866_

Round 1
Reviewer 1 Report
This is a very clearly written paper which makes good use of population-based survey data.
The obvious limitations of self-report data have been discussed, however, a consideration of the limitations of dichotomising outcome variables (loss of information as they were originally on a scale) should be included. Also, the crude measurement tools by which movement behaviour information was collected should be acknowledged.
Sleep guidelines usually have an upper limit - too much sleep could be associated with poorer mental health. This has not been addressed at all.
Only 15% met the physical activity guidelines - did you consider repeating the analysis only using the 60 min/d and not the strengthening guideline? Although incorporating both PA guidelines is a strength of this paper (and often overlooked in other papers), knowing what constitutes a muscle-strengthening activity may be a limitation to students answering this question, whereas simply reporting the time spent active may be easier.
Seems odd that so many participants met the screen guidelines. In the Canadian COMPASS survey, only 6.4% of grade 9 students met the screen guidelines. https://www.mdpi.com/1660-4601/17/15/5326
Among other adolescent populations, screen guideline compliance is similarly low. Please elaborate why compliance may be so unexpectedly high in this study.
Please add statistics about age to demographic Table.
A few grammar/typos:
on the Figure: X-axes – 1 behaviour (should be singular)
"accumulative relationships were primarily driven by meeting.." p 8 line 190
Suggest putting the "based on self-report" first (before average BMI) in this segment: "The average body mass index (BMI) was 23.6 ± 5.2 kg/m2. Based on self-report height and weight and the 2000 Centers for Disease Control and Prevention (CDC) BMI-for-age growth charts, the average BMI z-score was 0.61 ± 0.96."
Author Response
Reviewer #1:
The obvious limitations of self-report data have been discussed, however, a consideration of the limitations of dichotomising outcome variables (loss of information as they were originally on a scale) should be included. Also, the crude measurement tools by which movement behaviour information was collected should be acknowledged. -Thank you for this comment. We now acknowledge these limitations within the Discussion section (lines 335-338).
Sleep guidelines usually have an upper limit - too much sleep could be associated with poorer mental health. This has not been addressed at all. -Thank you. This limitation also has been commented on within the Discussion section (Lines 339-341).
Only 15% met the physical activity guidelines - did you consider repeating the analysis only using the 60 min/d and not the strengthening guideline? Although incorporating both PA guidelines is a strength of this paper (and often overlooked in other papers), knowing what constitutes a muscle-strengthening activity may be a limitation to students answering this question, whereas simply reporting the time spent active may be easier. -Thank you for this comment. We did run the analysis using just physical activity and the results were not statistically different compared to combining meeting both physical activity and muscle strengthening recommendations. However, we now added physical activity only descriptive statistics within Table 2 to provide data on the two different approaches.
Seems odd that so many participants met the screen guidelines. In the Canadian COMPASS survey, only 6.4% of grade 9 students met the screen guidelines. https://www.mdpi.com/1660-4601/17/15/5326. Among other adolescent populations, screen guideline compliance is similarly low. Please elaborate why compliance may be so unexpectedly high in this study. -Thank you for the comment. We now expand on this important finding within the Discussion section, and the provide the suggested reference (lines 245-256).
Please add statistics about age to demographic Table. -Thank you. Age statistics have now been added in the demographic Table 1.
A few grammar/typos:
on the Figure: X-axes – 1 behaviour (should be singular) -Thank you. This change has been made within Figure 1a and Figure 1b.
"accumulative relationships were primarily driven by meeting.." p 8 line 190 -Thank you. This sentence has been slightly amended (Lines 212-214).
Suggest putting the "based on self-report" first (before average BMI) in this segment: "The average body mass index (BMI) was 23.6 ± 5.2 kg/m2. Based on self-report height and weight and the 2000 Centers for Disease Control and Prevention (CDC) BMI-for-age growth charts, the average BMI z-score was 0.61 ± 0.96." -Thank you for this comment. These sentences have been changed (lines 97-99).
Reviewer 2 Report
This study has considered variables that had not been taken into account and that are decisive in mental health in adolescents, further clarifying the state of the question. The sample size endows the study with great rigor when discussing the results, and the recommendations after the results found are highly relevant to take into account to improve the well-being and mental health of adolescents.
Some of the suggestions to review at work are listed below:
Abstract
The authors do not place the reader in a rigorous conceptual framework, as there are many studies on physical activity and mental health. What is new about this type of analysis from the conceptual framework? Clear briefly. Point out the meaning of “multiple movement-based behaviors”
Why were these variables chosen and not others? What relationship do they have between them? "Associations between movement-based behaviors consisting of meeting 16 recommendations for physical activity, screen use, and sleep duration." Justify in the text.
The generalization of the evaluated variables to mental health should be done with more caution.
Introduction
This section can be developed with greater explanatory rigor and clarity, in relation to the variables under study. Explain in more detail and separately the relationship of each of the variables under study.
Line 61 "Feelings such as loneliness and sadness can contribute to mental health", justify this type of statement with scientific studies to support it.
Discussion
Line 231-236. Clarify in more detail what the results found may be due to.
Line 246-262. Clarify the scientific literary writing in relation to the variables studied. It becomes difficult to read the justification of the results
Conclusion
Line 305. In my opinion, highlighting and specifying again the dependent effect of the variables is extremely relevant in the conclusions.
This study has considered variables that had not been taken into account and that are decisive in mental health in adolescents, further clarifying the state of the question. The sample size endows the study with great rigor when discussing the results, and the recommendations after the results found are highly relevant to take into account to improve the well-being and mental health of adolescents.
Author Response
Reviewer #2:
Abstract
The authors do not place the reader in a rigorous conceptual framework, as there are many studies on physical activity and mental health. What is new about this type of analysis from the conceptual framework? Clear briefly. Point out the meaning of “multiple movement-based behaviors” -Thank you for this comment. The abstract has been amended to provide additional background and to provide needed clarification (lines 11-13).
Why were these variables chosen and not others? What relationship do they have between them? "Associations between movement-based behaviors consisting of meeting recommendations for physical activity, screen use, and sleep duration." Justify in the text. -Thank you for this comment. This has been clarified within the Abstract’s text (lines 11-13).
The generalization of the evaluated variables to mental health should be done with more caution. -Thank you for this comment. The concluding sentence of the abstract has been amended to be more specific with the generalizations (lines 24-26).
Introduction
This section can be developed with greater explanatory rigor and clarity, in relation to the variables under study. Explain in more detail and separately the relationship of each of the variables under study. -Thank you for this comment. The Introduction has now been expanded to discuss these behaviors individually and their potential impact on adolescent mental health (lines 32-40, lines 52-68, lines 80-87).
Line 61 "Feelings such as loneliness and sadness can contribute to mental health", justify this type of statement with scientific studies to support it. -Thank you for this comment. Two additional references have been provided (lines 80-84).
Discussion
Line 231-236. Clarify in more detail what the results found may be due to. -Thank you for the comment. More detail on the reasons for this finding have now been provided (lines 270-272).
Line 246-262. Clarify the scientific literary writing in relation to the variables studied. It becomes difficult to read the justification of the result. -Thank you for this comment. The writing within this paragraph has been amended to yield better clarity (lines 284-301).
Conclusion
Line 305. In my opinion, highlighting and specifying again the dependent effect of the variables is extremely relevant in the conclusions. -Thank you for this comment. Information regarding the observed joint associations have now been provided within the concluding paragraph (lines 349-351).
Reviewer 3 Report
Congratulations to the authors, the work is solid and interesting. I have only detected a small aspect to improve in Line 211, I think there should be a comma or a separation "928, 12 to 13 years-old"
Author Response
Reviewer #3:
Congratulations to the authors, the work is solid and interesting. I have only detected a small aspect to improve in Line 211, I think there should be a comma or a separation "928, 12 to 13 years-old" -Thank you for noticing this. Change made, line 234.